# Rethinking 3D Convolution in $\ell_p$-norm Space

**Li Zhang**[1,2,4], **Yan Zhong**[3], **Jianan Wang**[4], **Zhe Min**[5], **Rujing Wang**[1,2], **Liu Liu**[6*]

1 Hefei Institute of Physical Science, Chinese Academy of Sciences
2 University of Science and Technology of China, Hefei, China
3 School of Mathematical Sciences, Peking University. Beijing, China
4 Astribot, Shenzhen, China
5 Shandong University, Jinan, China
6 Hefei University of Technology, Hefei, China
zanly@mail.ustc.edu.cn, zhongyan@stu.pku.edu.cn

## Abstract

Convolution is a fundamental operation in the 3D backbone. However, under certain conditions, the feature extraction ability of traditional convolution methods may be weakened. In this paper, we introduce a new convolution method based on $\ell_p$-norm. For theoretical support, we prove the universal approximation theorem for $\ell_p$-norm based convolution, and analyze the robustness and feasibility of $\ell_p$-norms in 3D point cloud tasks. Concretely, $\ell_\infty$-norm based convolution is prone to feature loss. $\ell_2$-norm based convolution is essentially a linear transformation of the traditional convolution. $\ell_1$-norm based convolution is an economical and effective feature extractor. We propose customized optimization strategies to accelerate the training process of $\ell_1$-norm based Nets and enhance the performance. Besides, a theoretical guarantee is given for the convergence by *regret* argument. We apply our methods to classic networks and conduct related experiments. Experimental results indicate that our approach exhibits competitive performance with traditional CNNs, with lower energy consumption and instruction latency.

## 1 Introduction

The convolution-based 3D backbone networks have demonstrated substantial success in foundational tasks such as classification [1], object tracking [2], scene segmentation [3], etc. Some downstream tasks also heavily rely on these networks, such as interactive perception [4], object manipulation [5], imitation learning [6], and human-machine collaboration [7]. In the traditional 3D convolution, suppose $K \in \mathbb{R}^{m \times n}$ is the filter, and $P_t \in \mathbb{R}^{m \times n}$ is the sampled matrix from the $t$-th sliding window on input data, $1 \le t \le T$. $T$ is the total sliding counts. For any $t \ge 1$, the $t$-th convolution is calculated as:

$$P_t \odot K = \sum_{1 \le i \le m} \sum_{1 \le j \le n} P_t(i,j) \cdot K(i,j) \tag{1}$$

which is the same as inner product between vectors. To distinguish it from our new convolution framework, we refer to it as *inner product based convolution* in the following discussion. A geometric consideration arises when $P_t$ follows a certain symmetric distribution, such as a Gaussian or uniform distribution. By symmetry, there exist some of $\{P_t\}_{t=1}^T$ situated close to the subspace perpendicular to $K$, which means $K \odot P_t \approx 0$. This inevitably leads to explicit feature loss, diminishing the model's ability on information extraction.

---

*Li Zhang and Yan Zhong contribute equally. This work was done when Li Zhang was an intern at Astribot. Corresponding author: Liu Liu.

38th Conference on Neural Information Processing Systems (NeurIPS 2024).

In previous works, $\ell_p$-norms ($p = 1, 2, 3, \cdots, \infty$) demonstrated strong performance across various domains [8, 9, 10]. These norms exhibit remarkable capabilities in expressing spatial structures and local relationships within sets of points. To address the limitations of inner product-based convolution in certain extreme cases and to explore the potential of $\ell_p$-norms in feature extraction, we propose $\ell_p$-norm-based convolution, *i.e.*, for any kernel $K$ and sampled matrix $P_t$, it can be formulated as Eq. 2:

$$\|P_t - K\|_p \triangleq \big( \sum_{1 \leq i \leq m} \sum_{1 \leq j \leq n} (P_t(i,j) - K(i,j))^p \big)^{1/p}. \tag{2}$$

More precisely, the goal of this paper is to leverage the power of $\ell_p$-norm measurement (Fig. 1 (a)) and devise efficient and robust optimization methods for it. Our solutions are as follows:

From the theoretical standpoint, we prove the universal approximation theorem of $\ell_p$-norm Nets (for $p = 1, 2, 3, \cdots, \infty$). Besides, we show that $\ell_p$-norm based convolutions are more robust than the traditional ones via variance analysis under random noise.

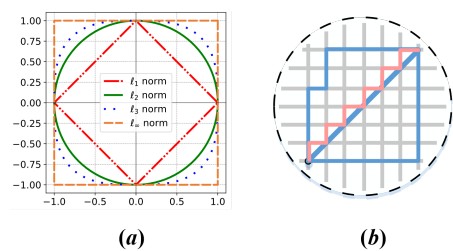

*(a)*        *(b)*

Figure 1: (**a**) Visualizing the circles of $\ell_p$-norms. (**b**) Manhattan distance based $\ell_1$-norm measure.

From the practical standpoint, we first discuss the performance of different $\ell_p$-norms in actual execution. 3D convolution in $\ell_\infty$-norm space tends to lose multiple useful pieces of information since only the maximum absolute value is reserved. The $\ell_2$-norm measure is inherently a linear transformation of the traditional convolution (details can be found in Sec. A). In contrast, the $\ell_1$-norm has unique potential for 3D point cloud tasks. However, directly replacing traditional convolution with an $\ell_1$-norm-based one is not feasible in practice due to the difficult convergence and local optima. To enhance network performance, we propose customized optimization strategies. The first strategy is a mixed gradient strategy (MGS), and the second is a dynamic learning rate controller (DLC). These strategies are applied in the training process (Algorithm 1) to accelerate network convergence and avoid local optima. We also provide a convergence guarantee for our optimization strategies from the perspective of *regret*.

We evaluate our method on several benchmarks, ranging from global, semi-dense, and dense prediction tasks. The experimental results show that $\ell_1$-norm Net has the same competitive performance as traditional convolution. Moreover, the proposed $\ell_1$-norm Net has three advantages: 1) $\ell_1$-norm (inherently addition operation) has lower computational complexity compared to multiplication; 2) addition significantly reduces energy consumption [11]; 3) $\ell_1$-norm operations (addition) has lower instruction latencies [12] than inner product process (multiplication). These properties facilitate the 3D point cloud tasks especially online tasks such as 3D real-time object detection, pose tracking, etc.

**Contributions.** 1) We prove the universal approximation for $\ell_p$-norm Nets. And we show that $\ell_p$-norm Nets are robust under random noise. 2) We compare different $\ell_p$-norm based convolutions, and further propose a reliable and efficient $\ell_1$-norm Net for 3D point cloud tasks with customized optimization strategies. We also give a theoretical guarantee for convergence by regret argument. 3) Experimental results demonstrate the effectiveness of our methods in 3D point cloud tasks, showing lower energy consumption and faster instruction execution.

## 2 Related Work

**Different Convolution Methods.** Convolutions have seen significant success, leading to various convolution methods aimed at improving performance and efficiency. Traditional convolutions, introduced by [13], use fixed-size kernels to extract features but are computationally intensive and may not capture diverse patterns effectively. To overcome these limitations, several alternatives have been proposed: 1) depthwise separable convolutions [14, 15]. Popularized by MobileNets, these decompose standard convolutions into depthwise and pointwise operations. 2) dilated convolutions [16, 17, 18]. These introduce spaces between kernel elements, expanding the receptive field without increasing parameters. 3) deformable convolutions [19, 20]. These adapt the sampling loca-

tions of the convolutional kernel, enhancing the network's ability to model geometric transformations. However, due to their unique strengths, they only excel at some specific tasks.

$\ell_p$-**norm Measure in Different Tasks.** Using the $\ell_p$-norm as a feature measurement function for convolutional kernels offers several advantages: 1) Flexibility: The $\ell_p$-norm allows adjusting the parameter $p$ according to specific needs [21, 22, 23]. 2) Sparsity: It encourages most elements in the convolutional kernel to approach zero, reducing computational complexity and storage requirements [21, 24]. Overall, in diverse settings, employ distinctive approaches. The $\ell_p$-norm is widely used across various fields. For example, in *image processing*, the $\ell_1$-norm is used for sparse representation in image compression [25], enabling efficient storage and transmission. In *machine learning and optimization*, optimization problems also use $\ell_p$-norm constraints to impose sparsity or specific patterns in solutions [26, 27]. Despite progress, directly migrating these methods into 3D point cloud tasks causes a *domain gap*. In this work, we aim to explore $\ell_p$-norm measure for 3D point cloud tasks in depth.

## 3 Methodology

**Notations.** For the sake of simplicity, in what follows, we take the classic PointNet++ [28] as the basis model to estimate the efficiency of $\ell_p$-norm based Nets with the proposed optimization strategies. Note that, we directly replace the inner product based convolution by $\ell_p$-norms ($p = 1, 2, 3, \ldots, \infty$) based one, and denote the corresponding network by $\ell_p$-PointNet++ or $\ell_p$-norm Net. Moreover, the proposed $\ell_p$-norm based convolution can also be called $\ell_p$-norm neuron.

### 3.1 Universal Approximation

The universal approximation ability of a neural network is crucial. Firstly, it establishes a solid theoretical foundation for the network's capabilities [29], which asserts that certain architectures and activation functions enable neural networks to approximate any continuous function. There is a series of works on the approximation capacity, such as theories for feedforward networks [30], RNNs [31], Transformer [32]. However, the universal approximation property of $\ell_p$-PointNet++ has not been studied thoroughly up to now.

**Theorem 1.** *Assume $S = \{x_1, \cdots, x_N\} \subset \mathbb{R}^k$ is an arbitrary point cloud. $J \subset \mathbb{R}^k$ is any compact set and $S \subset J$. For any continuous function $f$ defined on $2^J$ with respect to Hausdroff distance $d_H(\cdot, \cdot)$, there exists an $\ell_p$-PointNet++ $\mathcal{P}$ satisfying for any $\epsilon > 0$,*

$$|f(S) - \mathcal{P}(S)| \leq \epsilon. \tag{3}$$

*Moreover, for any $\ell_1$-integrable function $g$ defined on $J$, there exists an $\ell_p$-PointNet++ $\mathcal{P}'$, for any $\epsilon' > 0$,*

$$\int_{x \in J} |g(x) - \mathcal{P}'(x)| dx < \epsilon'. \tag{4}$$

Briefly speaking, $f$ could be approximated by an MLP consisting of $\ell_p$-norm convolution layers and a max pooling layer. And $g$ could be approximated by a network composed of an $\ell_p$-norm convolution layer and a fully connected layer. The detailed proof can be found in Sec. A from the appendix.

### 3.2 Robustness Analysis

In the following, we show that under Gaussian random noise on input data, $\ell_p$-norm based convolutions are more robust than that based on inner product. Suppose $G \in \mathbb{R}^{m \times n}$ is a Gaussian matrix. Each $G(i, j) \sim N(0, \sigma^2)$ where $\sigma > 0$ is a constant. Let $P_t \in \mathbb{R}^{m \times n}$ be the data at time $t$ and $K \in \mathbb{R}^{m \times n}$ be the kernel function.

For inner product,

$$Var\big[(G + P_t) \odot K\big] = \mathbb{E}_G\Big[\big(G \odot K - \mathbb{E}_G[G \odot K]\big)^2\Big] = Var\big[G \odot K\big], \tag{5}$$

and

$$G \odot K = \sum_{i=1}^{m} \sum_{j=1}^{n} G(i, j) K(i, j) \sim N\bigg(0, \sigma^2 \cdot \sum_{i=1}^{m} \sum_{j=1}^{n} K(i, j)^2\bigg). \tag{6}$$

Suppose $\forall i \in [m]$ and $\forall j \in [n]$, $K(i,j)$ is a constant, we have $Var\big[(G + P_t) \odot K\big] = \Theta(mn)$.

For $\ell_p$-norm, first we could prove that when $p = 2$, $Var\big[\|G + X - K\|_2\big] = O(1)$, which is significantly smaller than $Var\big[(G + P_t) \odot K\big]$. The details of calculation could be found in Sec. A from the appendix. For the more general cases ($p = 1, 2, 3, \cdots, \infty$), we show that $\ell_p$-norm has a small variance through numerical computation in the Tab. 1, where we take $\sigma = 1$.

Table 1: **Variance of the $\ell_p$-norm of Gaussian random vector when $mn = 9$.**

| p | 1 | 2 | 3 | 4 | 5 |
|---|---|---|---|---|---|
| Var | 3.24655 | 0.48327 | 0.31248 | 0.27494 | 0.26078 |
| p | 6 | 7 | 8 | 9 | $\infty$ |
| Var | 0.26093 | 0.26040 | 0.26078 | 0.26145 | 0.26875 |

### 3.3  Implementation of $\ell_p$-norm Nets

Note that although Theorem 1 guarantees a universal approximation capability, it does not mean that all the $\ell_p$-norm Nets are efficient and feasible in practice. Therefore, we further discuss the characteristics of each specific $\ell_p$-norm Nets ($p = 1, 2, 3, \cdots, \infty$) in detail.

Assume the input data follows Gaussian distribution, saying $G$ is the standard Gaussian matrix. For $\ell_p$-norm based convolution, when $p$ is greater than or equal to 3, the distribution of the output data is very close. We present the simulation results in Fig. 2. It's clear that when $p$ is getting larger, the distribution of $\|G\|_p$ gradually overlaps with the distribution of $\|G\|_\infty$. Therefore, we take $p = \infty$ as the representative case for $p \geq 3$.

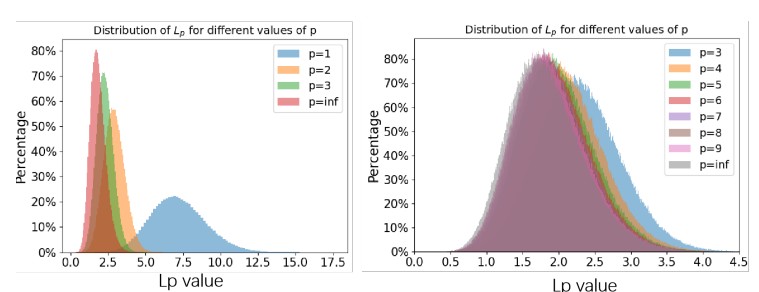

Figure 2: (**Left**) The distribution of $\|G\|_p$, where $G$ is the standard Gaussian vector, $p = 1, 2, 3, \infty$ and $dim(G) = 9$. (**Right**) The distribution of $\|G\|_p$, $p = 3, 4, 5, 6, 7, 8, 9, \infty$ and $dim(G) = 9$.

Actually, $l_\infty$-norm exhibits weaknesses due to its overly simplistic emphasis on the largest element. Namely this approach tends to oversimplify the feature space by disproportionately emphasizing only one dimension, potentially discarding valuable information present in other dimensions. Also, this concept is supported by experimental results in Sec. 5. Besides, $\ell_2$-norm inherently is calculated by taking the square root of the sum of the squares of its elements. And $\ell_2$-norm based convolution $\mathcal{C}_{\ell_2}$ can be regarded as an equivalence transformation of the traditional convolution $\mathcal{C}$. Briefly speaking, we could show that $\mathcal{C}_{\ell_2}^2 = \alpha + \beta \times \mathcal{C}$, where $\alpha$ and $\beta$ are constants.

$\ell_1$-norm can synthesize each element of the feature vector. And the $\ell_1$-norm Net is not equivalent to a translation transform, which we believe holds potential as a 3D convolutional similarity metric function according to the Theorem. 1. To this end, our method focuses on rationalizing the $\ell_1$-norm measure to maximize its potential in feature extraction. Mathematically, if the similarity measurement function between the input data and kernel function is replaced with the $\ell_1$-norm, the convolution can be re-formulated as:

$$Y(P_t, K) = -\sum_{t \geq 1} \sum_{i,j} |P_t(i,j) - K(i,j)| \tag{7}$$

The underlying operation of $\ell_1$-norm kernel function is addition, which has more development potential and application value in real scenarios. Specifically, 1) It contains almost no multiplication but addition, resulting in lower computational complexity of the model. 2) $\ell_1$-norm operation (addition) is proved to have lower energy consumption compared to the inner product (multiplication) calculation [33]. Take the operation of floating-point addition and multiplication as an example, which has energy costs of 0.9 $pJ$ and 3.7 $pJ$, respectively. 3) Low latency is also a consideration in practical application scenarios. [12] tells us that multiplication (inner product process) has longer theoretical instruction wait times than addition operations. Table 1 of this study lists the instruction

latency, throughput, and micromanipulation faults for Intel, AMD, and VIA CPUs. For instance, the latency of float multiplication and addition is 4 and 2 in the VIA Nano 2000 series.

## 3.4 Regret

It's a good way [34, 35, 36] to demonstrate the convergence of an optimization process by analyzing the *regret*. Performance measurement [37], optimization guidance [38], and feedback mechanisms [39] can be summarized as its advantages. We employ it the construct the convergence theorem for our optimization strategies in Sec. 4.

Consider a general online optimization model between a player and an adversary. A subset $\mathcal{F} \in \mathbb{R}^m$ is non-empty, bounded and closed. For each iteration $k \in [T^*]$, the player choose a point $\mathbf{x}_k \in \mathcal{F}$ ($T^*$ is not known for player). After committing to this choice, a convex function $h_k$ will be revealed by the adversary. And we note the cost of this game by *regret*:

$$R_{T^*} = \sum_{k=1}^{T^*} h_k(\mathbf{x}_k) - \min_{\mathbf{x} \in \mathcal{F}} \sum_{k=1}^{T^*} h_k(\mathbf{x}). \tag{8}$$

The player aims to carefully select $\mathbf{x}_k$ to minimize regret as much as possible, while conversely the adversary aims to specifically choose $h_k$ to hinder the player. Intuitively, if an algorithm(the player) could bound regret by a sub-linear function of $T^*$, *i.e.*, $R_{T^*} = o(T^*)$, we could conclude that "on the average" the algorithm performs as well as the best fixed strategy in hindsight [40].

## 4 Optimization

By the argument above, we are motivated to devise a new convolution based on $\ell_1$-norm. However, direct training of $\ell_1$-norm Nets can easily lead to unsatisfactory results. Thus, two customized optimization strategies are proposed for training. Before introducing these optimization strategies, we clarify the notations in the following.

**Notations**  Recall that $K \in \mathbb{R}^{m \times n}$ is the kernel and $P_t \in \mathbb{R}^{m \times n}$ is the sliding window on the input data, $1 \leq t \leq T$. $Y(P_t, K)$ is the convolution of $K$ and $P_t$. $L$ denotes the loss function in training process. We use the $m \times n$ matrix $\frac{\partial L}{\partial K}$ to denote the gradient on of $L$ on $K$, where $(\frac{\partial L}{\partial K})_{i,j} = \frac{\partial L}{\partial K(i,j)}$. Besides, define the vectors

$$\frac{\partial L}{\partial Y} \triangleq \Big( \frac{\partial L}{\partial Y(P_1, K)}, \frac{\partial L}{\partial Y(P_2, K)}, \ldots, \frac{\partial L}{\partial Y(P_T, K)} \Big)$$

and

$$\frac{\partial Y}{\partial K(i,j)} \triangleq \Big( \frac{\partial Y(P_1, K)}{\partial K(i,j)}, \frac{\partial Y(P_2, K)}{\partial K(i,j)}, \ldots, \frac{\partial Y(P_T, K)}{\partial K(i,j)} \Big)$$

### 4.1 MGS: Mixed Gradient Strategy

Now we focus on the gradient descent in training process, especially the partial derivative of loss function $L$ on the kernel $K$. It should be pointed out that $L$ is a function on $(Y(P_1, K), Y(P_2, K), \ldots, Y(P_T, K))$. By chain rule of derivation we have for any given $(i, j)$,

$$\frac{\partial L}{\partial K(i,j)} = \sum_{t=1}^{T} \frac{\partial L}{\partial Y(P_t, K)} \cdot \frac{\partial Y(P_t, K)}{\partial K(i,j)} = \Big\langle \frac{\partial L}{\partial Y}, \frac{\partial Y}{\partial K(i,j)} \Big\rangle \tag{9}$$

Notice that when loss function $L$ is fixed, $\frac{\partial L}{\partial Y}$ is regardless of the choice of $Y(P_t, K)$ (inner product or $\ell_p$-norm). And we should only focus on the vector $\frac{\partial Y}{\partial K(i,j)}$. In the context of $\ell_1$-PointNet++:

$$\frac{\partial Y(P_t, K)}{\partial K(i,j)} = \text{sgn}\big(P_t(i,j) - K(i,j)\big). \tag{10}$$

Here, $\text{sgn}(\cdot)$ represents the sign function.

There are two unavoidable problems: 1) the use of Eq. 10 results in a signSGD update. As discussed in [41], the direction of signSGD is not aligned with the steepest descent, and this misalignment exacerbates with increasing dimensionality. 2) The gradient of $\ell_1$-norm Net is significantly smaller than that of inner product convolution in the experiment. Namely, $\|\frac{\partial L}{\partial K}\|_2$ is extremely small when we choose the convolution $Y$ as $\ell_1$-norm. Taking PointNet++ on S3DIS as an example, we report the $\ell_2$ norm of gradient of $\ell_1$-PointNet++ in Fig. 3. The gradient from $\ell_1$-PointNet++ is much smaller than that in PointNet++ (*e.g.*, $\ell_1$-PointNet++: 0.0002, PointNet++: 0.3162 in layer I). Hence, this small gradient $\frac{\partial L}{\partial K}$ in $\ell_1$-norm Net would significantly slow down the training process.

Based on the above observations, we introduce a novel **M**ixed **G**radient **S**trategy (**MGS**) tailored for $\ell_1$-PointNet++ training. This approach strategically combines the gradients of the $\ell_1$-PointNet++ and that of $\ell_2$-PointNet++:

$$\frac{\partial Y(P_t, K)}{\partial K(i,j)} = \frac{P_t(i,j) - K(i,j)}{||K - P_t||_2}, \quad (11)$$

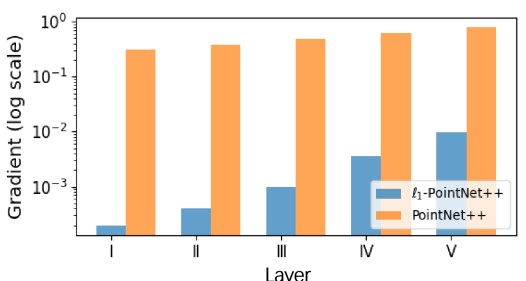

Actually, as we discussed above, $\ell_2$-norm based convolution is a linear transform of inner product convolution. So gradient of $\ell_2$-norm Net has a proper scale. The mixed strategy involves dynamically adjusting $\frac{\partial Y(P_t,K)}{\partial K(i,j)}$ during training, guided by a parameter $0 < \lambda < 1$ and the training step $k$. **The mixed gradient strategy** is expressed as:

Figure 3: **The gradient of weight in each layer using two different networks at 1st iteration.** Layer I to III represent 3 SetAbstractions modules in $\ell_1$-PointNet++ and layer IV to V represent fully connected layers. Note that the y-axis is on a logarithmic scale to reflect the magnitude of the values.

$$\frac{\partial Y(P_t, K)}{\partial K(i,j)} = (1 - \lambda^k)\mathrm{sgn}(P_t(i,j) - K(i,j)) + \lambda^k(P_t(i,j) - K(i,j)). \qquad (12)$$

This dynamic adjustment introduces a controlled transition in the gradient computation as training progresses. Taking $\lambda = 0.99$ for example, when $k$ is small, the term $\lambda^k$ dominates and $\frac{\partial Y(P_t,K)}{\partial K(i,j)}$ approximates to $P_t(i,j) - K(i,j)$. This initial configuration aligns with the more efficient $\ell_2$-like update, providing stability and aiding in faster convergence. As training progresses ($k$ gets larger), the term $\lambda^k$ becomes more prominent, shifting the gradient computation towards $\mathrm{sgn}(P_t(i,j) - K(i,j))$. This transition allows the model to leverage the advantages of the $\ell_1$-PointNet++ structure, facilitating sparsity in the learned features. By dynamically adapting the gradient computation based on the training step, the mixed strategy offers a flexible and adaptive approach to overcome the challenges associated with fixed gradient schemes. This dynamic adjustment provides a thoughtful compromise, combining the efficiency of $\ell_2$-like updates in the initial stages with the sparsity-inducing benefits of $\ell_1$-PointNet++ in later stages.

In fact, there is quite a bit of literature supporting the effectiveness of the signSGD update scheme, and in particular, it has been shown that it has some advantages in avoiding saddle points [42]. However, when certain random rotations of the objective appear, signSGD may become trapped in a periodic behavior that hinders convergence in such cases. To address this unexpected behavior, we additionally explored the introduction of momentum into the update rule. Our experimental results prove that this modification effectively breaks the symmetry induced by random rotations, preventing the model from getting stuck and fostering smoother convergence.

## 4.2 DLC: Dynamic Learning rate Controller

Considering the uniqueness of the mixed gradient strategy, we focus on achieving larger update magnitudes and faster convergence rates during the initial stages of training. However, in the later stages, we aim to revert to signSGD, implementing a more cautious update strategy to enhance the model's precision. Therefore, we propose a learning rate update strategy that adapts to this characteristic: **D**ynamic **L**earning rate **C**ontroller (**DLC**), maintaining a higher rate in the early training phase, and returning to a lower rate in the later phase.

To this end, we design two bound functions to control the learning rate: the lower bound

$$\alpha_1(k) = p_1 \cdot (1 + \frac{p_2}{e^k}) \tag{13}$$

and the upper bound

$$\alpha_2(k) = p_1 \cdot (1 + \frac{p_3}{k}) \tag{14}$$

where $p_1$, $p_2$ and $p_3$ are hyper-parameters to be determined and $k$ denotes the training step. And we use simple comparison operations to make learning rate $\alpha(k)$ locate in $[\alpha_1(k), \alpha_2(k)]$:

$$\hat{\alpha}(k) \leftarrow \min\{\max\{\alpha_1(k), \mathcal{A}[\alpha(k)]\}, \alpha_2(k)\}. \tag{15}$$

To enhance the universality of this dynamic control framework, $\mathcal{A}$ could be another learning rate optimization algorithm like the adaptive learning rate strategy of [43], which can be specifically switched according to the task at hand. However, regardless of $\mathcal{A}$, we will later demonstrate that dynamic control alone is sufficient to provide theoretical convergence guarantees by the regret argument of Theorem 2, and it also performs well in experiments.

### 4.3 Training Framework

It has been noted from previous discussions that the momentum method can help signSGD avoid getting trapped in cycles, thereby improving training stability. Combining the methods above, we present the global optimization algorithm (**O**ptimizer with **M**ixed gradient strategy and **D**ynamic learning rate controller, **OMD**) for $\ell_1$-PointNet++ training. Details are shown in Algorithm. 1

Here we give a convergence guarantee for **OMD** under an online optimization framework, which is harder than offline optimization. We could show that regret $R_{T^*}$ of **OMD** is bounded by $O(\sqrt{T^*})$. Low regret means the algorithm

---

**Algorithm 1** OMD

**Input:** Initial learning rate $\alpha$, hyper-parameters $p_1, p_2, p_3$, referred by Eq. 13 and Eq. 14. $q_0$ and $q$ in $(0,1)$.

1: $\mathbf{m}_0 = 0$, $\alpha(0) = \alpha$, $\mathbf{x}_1 = \vec{0}$.
2: Set the functions $\alpha_1(k)$ and $\alpha_2(k)$ by hyper-parameters $p_1, p_2, p_3$.
3: **for** $k = 1$ to $T^*$ **do**
4: $\quad \mathbf{g}_k \leftarrow \frac{\partial L}{\partial K}(\mathbf{x}_k)$     # Consider the gradient $\frac{\partial L}{\partial K}$ as an vector here. $\frac{\partial L}{\partial K(i,j)} = \langle \frac{\partial L}{\partial Y}, \frac{\partial Y}{\partial K(i,j)} \rangle$. $\frac{\partial L}{\partial Y}$ only depends on the choice of loss function. See Eq. 12 for $\frac{\partial Y}{\partial K(i,j)}$.
5: $\quad q_k = q_0 \cdot q^k$.
6: $\quad \mathbf{m}_k = q_k \cdot \mathbf{m}_{k-1} + (1 - q_k) \cdot \mathbf{g}_k$
7: $\quad \hat{\alpha}(k) \leftarrow \min\{\max\{\alpha_1(k), \mathcal{A}[\alpha(k)]\}, \alpha_2(k)\}$
8: $\quad \alpha(k) \leftarrow \hat{\alpha}(k)/\sqrt{k}$
9: $\quad \mathbf{x}_{k+1} = \Pi_{\mathcal{F}, \alpha(k)^{-1/2}}(\mathbf{x}_k - \alpha(k) \cdot \mathbf{m}_k)$
10: **end for**

---

progressively gets closer to the optimal solution over time. This shows that **OMD** has reliable convergence properties, making it a dependable optimization method.

**Theorem 2.** *Continue with the settings and notations of Algorithm 1. Suppose $\mathcal{F} \subset \mathbb{R}^n$ is bounded, saying $\max_{\mathbf{x},\mathbf{y} \in \mathcal{F}} \|\mathbf{x} - \mathbf{y}\|_\infty \leq B_\infty$ Besides, suppose $\forall k \in [T^*]$, $\|\mathbf{g}_k\|_2 \leq B_2$. we could show that for any convex functions $\{h_k\}_{t=1}^{T^*}$,*

$$R_{T^*} = \sum_{k=1}^{T^*} h_k(\mathbf{x}_k) - \sum_{t=1}^{T^*} h_k(\mathbf{x}^\star) \leq C_1 \cdot \sqrt{T^*} + C_2$$

*where $C_1$ and $C_2$ are constants that rely on $p_1$, $p_2$, $p_3$, $B_\infty$, $B_2$, $q_0$ and $q$. And $\mathbf{x}^\star \triangleq \arg\min_{\mathbf{x} \in \mathcal{F}} \sum_{k=1}^{T^*} h_k(\mathbf{x})$.*

The proof could be found in the appendix, Sec. A.

## 5 Experiments

To validate the generalizability and robustness of the method and thus ensure its effectiveness and broad applicability, we verify the performance of our method in several tasks, ranging from **Global Tasks** (*i.e.*, Parts Segmentation), **Semi-dense Prediction** (*i.e.*, scenario semantic segmentation), and **Dense Prediction** (*i.e.*, pose estimation) tasks. Shapenet, S3DIS, and GarmentNets Simulation are used as the datasets.

## 5.1 Dataset and Experimental Settings

**Dataset.** 1) ShapeNet. In ShapeNet, there are 16,881 shapes from 16 categories, which are annotated with 50 parts in total. Note that most object categories are labeled with two to five parts and Ground Truth annotations are labeled on sampled points on the shapes. This task can be regarded as a point-wise classification task. 2) S3DIS. The Stanford Large-Scale 3D Indoor Spaces Dataset, which encompasses 3D scans obtained from Matterport scanners across 6 distinct areas, comprising a total of 271 rooms. Within the S3DIS dataset, every point within the scans is labeled with a semantic category from a set of 13 distinct classes. These classes encompass various elements such as chairs, tables, floors, walls, among others, in addition to a category for clutter. 3) GarmentNets Simulation. GarmentNets Simulation is a large-scale dataset proposed by [44]. This dataset has six garment categories with a total data volume of 1.72TB. Dress, Jump, Skirt, Top, Pants and Shirt are included.

**Experimental Settings.** We train our frameworks using CrossEntropy loss and the AdamW optimizer [45], with an initial learning rate of 0.001, a weight decay of $10^{-4}$, Cosine Decay, and a batch size of 32. The total training consists of 200 epochs. *All tasks use the same settings unless otherwise specified.* All experiments are conducted on a computer workstation with three GeForce GTX 3090 GPUs using the PyTorch deep learning framework. The best model on the validation set is selected for testing.

## 5.2 Experiments on Global Task

**Parts Segmentation.** As a classic global task, 3D object parts segmentation is an important predecessor for articulated objects from the embodied intelligence community, such as pose estimation [46, 47], manipulation [48, 49], etc. In this section, we conduct experiments on ShapeNet part dataset [50].

Table 2: **Quantitative segmentation results on ShapeNet part dataset**. Note that a 3D fully convolutional network is proposed as the 3DCNN, mIoU(%) is reported as the metric on points.

| Model | Mean | Shape Names | | | | | | | | | | | | | | | |
|---|---|---|---|---|---|---|---|---|---|---|---|---|---|---|---|---|---|
| | | aero | bag | cap | car | chair | ear phone | guitar | knife | lamp | laptop | motor | mug | pistol | rocket | skate board | table |
| # shape number | | 2690 | 76 | 55 | 898 | 3758 | 69 | 787 | 392 | 1547 | 451 | 202 | 184 | 283 | 66 | 152 | 5271 |
| 3DCNN | 79.4 | 75.1 | 72.8 | 73.3 | 70.0 | 87.2 | 63.5 | 88.4 | 79.6 | 74.4 | 93.9 | 58.7 | 91.8 | 76.4 | 51.2 | 65.3 | 77.1 |
| $\ell_1$-3DCNN | **79.4** | 79.3 | 70.9 | 71.3 | 72.9 | 86.3 | 58.6 | 90.0 | 76.5 | 74.9 | 92.6 | 63.8 | 89.9 | 75.8 | 55.8 | 63.6 | 79.2 |
| PointNet | **83.7** | 83.4 | 78.7 | 82.5 | 74.9 | 89.6 | 73.0 | 91.5 | 85.9 | 80.8 | 95.3 | 65.2 | 93.0 | 81.2 | 57.9 | 72.8 | 80.6 |
| $\ell_1$-PointNet | 83.3 | 85.9 | 76.5 | 78.3 | 78.3 | 86.1 | 75.6 | 89.8 | 85.3 | 81.0 | 97.7 | 62.3 | 91.2 | 83.9 | 60.1 | 73.0 | 80.7 |
| PointNet++ | 86.2 | 87.1 | 80.3 | 86.3 | 74.3 | 90.1 | 75.3 | 92.9 | 86.3 | 79.3 | 94.9 | 71.2 | 93.3 | 81.9 | 59.2 | 73.8 | 81.2 |
| $\ell_1$-PointNet++ | **86.5** | 89.1 | 78.3 | 85.2 | 76.3 | 86.9 | 74.6 | 93.1 | 84.9 | 79.6 | 94.6 | 72.6 | 91.3 | 81.3 | 66.3 | 74.8 | 81.8 |

From Tab. 2, see them all: we find that our method has almost equivalent performance to the conventional method when being equipped with PointNet, and achieves superior performance on 3DCNN and PointNet++. Treat them equally: we see that our method can often perform better in some categories (*e.g.*, car, motor, rocket, etc.), these categories usually have a larger volume (*i.e.*, a more sparsified point cloud) compared to other objects. We propose that the inner product within convolutional networks has a tendency to highlight local context among points, yet it is greatly affected by the overall translation and scaling of the dataset. Our method focuses on the points drawn from $\ell_1$-norm space and addresses this problem by integrating the inherent distance measure into our architecture. Specifically speaking, The Manhattan distance based $\ell_1$-norm Nets tend to avoid this problem, which notices point cloud features at a longer distance.

## 5.3 Experiments on Semi-dense Prediction Task

**Scenario Semantic Segmentation.** As a semi-dense prediction task, this task aims to segment distinct regions within a 3D scene based on their semantic meaning using point cloud data. Semantic scene segmentation is crucial for understanding and interpreting the spatial arrangement and relationships between objects in 3D scenes. For our study, we utilize the S3DIS dataset. The metrics and experimental settings follow those outlined in [28].

Following the training and test strategies used in [51], we first divide the point cloud using the room as the basic unit and then sample the room at a size of 1m * 1m (randomly sampling up to 4096

Table 3: **(Left) Results on Semantic Segmentation in Scenes.** Metric is average IoU (%) over 13 classes (structural and furniture elements plus clutter) and classification accuracy is calculated on points. Our methods achieved competitive performance with significant energy reductions (**61%**).



| Model | Mean IoU (%) | Overall Accuracy (%) | Energy ($\mu j$) |
|---|---|---|---|
| PointNet | **47.7** | **78.6** | 7.981 |
| $\ell_1$-PointNet | 47.6 | 77.9 | **3.471** |
| PointNet++ | 53.5 | **83.0** | 3.395 |
| $\ell_1$-PointNet++ | **53.9** | 82.9 | **1.328** |

*(a) Input*

*(b) Output*

Figure 4: **(Right) Qualitative Results for Semantic Segmentation.** We put the colored point cloud on the top part (Input data) and put semantic segmentation results from the same camera viewpoint on points (Output) in the bottom part.

points during training, and all points are involved in the computation during testing), which in turn predicts the class of each point in each block. Note that we use a 9-dimensional vector to represent each point, representing XYZ, RGB, and normalized room location (ranging from 0 to 1). K-fold strategy is also used for training and testing.

The quantitative results are reported in Tab. 3. Experimental results show that although our approach achieves almost equivalent performance to inner product based networks, we maximize the potential of $\ell_1$-norm measure by relying on our proposed optimization strategy, which allows us to achieve similar performance but with less computational complexity and lower energy consumption (Almost **61%** energy reductions). Also, we provide qualitative segmentation results for visualization in Fig. 4. Overall, our model generates consistent object predictions and is resilient to the presence of absent points and obstructions.

## 5.4 Experiments of Dense Prediction Task

**Garment Pose Estimation** Garments, vital in daily life, present unique challenges for machine perception and interaction due to properties like infinite degrees of freedom and thin structure. Garment pose estimation and tracking systems hold potential for applications in mixed reality [52, 53], augmented reality [54, 52], and robotic manipulation [55, 49]. Addressing these challenges, mainstream methods typically employ Normalized Object Coordinate Space (NOCS) [56] for **dense prediction tasks**. In this section, we introduce GarmentNets [44], a baseline focusing on garment pose estimation using partial point clouds as input and generating complete point clouds as output. Our approach utilizes the GarmentNets Simulation Dataset to evaluate this task. The total epoch number is 200, and the batch size is 16.

Table 4: **Quantitative Results on Garment Pose Estimation.** The metric is measured using Chamfer distance ($cm$) under the canonical pose. The lower is the better result.

| Model | Dress | Jumpsuit | Skirt | Top | Pants | Shirt |
|---|---|---|---|---|---|---|
| GarmentNets [44] | 1.94 | **1.45** | 2.00 | 1.30 | 1.03 | 1.70 |
| $\ell_1$-GarmentNets | **1.83** | 1.56 | **1.91** | **1.26** | **0.99** | **1.62** |

Quantitative results are in Tab. 4. Note that we use Symmetric Chamfer Distance as the metric, This metric measures accuracy and completeness for surface reconstruction. The accuracy metric is defined as the mean L2 distance of points on the output mesh to their nearest neighbors on the GT mesh. From the table, it can be seen that our method performs comparably to the original method.

## 5.5 Ablation Experiments

**Replacing Means.** The most critical structure of PointNet++ is the 3 separate SetAbstractions modules (SA). Hence, to explore the effect of using the $\ell_1$-PointNet++ at different places and in different ratios, we remove the modules at different ratios and places on S3DIS. The experimental result is shown in Tab. 5. In many aspects, we can infer that the average mean IOU and accuracy are higher under the 66.7% ratio than those reported under the 33.3% ratio. This result tells the conclusion that our $\ell_1$-norm measure can exact more useful features from sparse point clouds. we hope these results can prompt further study on replacing means, such as different replacing ratios in

each inner module, creditable ways to combine hybrid convolutional blocks. etc. We leave this for more passionate researchers in the future.

Table 5: **Comparisons of Results on S3DIS with Different Replacing Ratio and Places**. We estimate the energy costs according to [11], *i.e.*, one operation of floating-point addition and multiplication have energy costs of 0.9 $pJ$ and 3.7 $pJ$, respectively. SA: SetAbstractions module. ★ means that 33.3% replacing ratio of PointNet++ has #Add-0.492 M, #Mul-0.984 M, Energy-4.0836 $\mu J$, while ♣ means that 66.7% replacing ratio of PointNet++ has #Add-0.984 M, #Mul-0.492 M, Energy-2.7060 $\mu J$.

| Replacing Ratio | $\ell_1$-norm neuron? | | | Mean IOU | Accuracy | Info |
|---|---|---|---|---|---|---|
| | SA1 | SA2 | SA3 | | | |
| 33.3% | ✓ | | | 51.9% | 79.8% | |
| | | ✓ | | 52.3% | 81.8% | ★ |
| | | | ✓ | 52.5% | 81.1% | |
| 66.7% | | ✓ | ✓ | 53.2% | 82.4% | |
| | ✓ | ✓ | | 53.0% | 81.0% | ♣ |
| | ✓ | ✓ | | 52.2% | 81.5% | |

Table 6: **Ablation Results on S3DIS Dataset Using Different Variants of $\ell_1$-Nets.** Mean IOU and overall Accuracy (%) are reported. Note that the results of $\ell_1$-PointNet are reported from I to IV, and $\ell_1$-PointNet++ are reported from V to VIII. Besides, vanilla Net represents the model without our customized optimization strategy while training.

| Index | Optimization? | | Mean IOU (%) | Overall Accuracy (%) |
|---|---|---|---|---|
| | MGS | DLC | | |
| I (Vanilla) | | | 33.2% | 56.3% |
| II | ✓ | | 39.6% | 68.6% |
| III | | ✓ | 42.8% | 70.1% |
| IV (Ours) | ✓ | ✓ | 47.6% | 77.9% |
| V (Vanilla) | | | 38.9% | 55.6% |
| VI | ✓ | | 43.6% | 69.3% |
| VII | | ✓ | 48.4% | 75.6% |
| VIII (Ours) | ✓ | ✓ | 53.9% | 82.9% |

**Optimization Strategy.** As demonstrated in Sec. 4, we propose mixed gradient strategy (MGS) to accelerate network convergence, while dynamic learning rate controller (DLC) helps our network move away from local optima. To evaluate the effectiveness of MGS and DLC, we remove them separately from $\ell_1$-PointNet++ and evaluate the scenario semantic segmentation performance on S3DIS. Tab. 6 presents the quantitative results. The baselines (I and V) indicate that we only use $\ell_1$-norm as the similarity measurement but without any optimization. It can be observed that they both resulted in huge performance degradation. Besides, we can see that both our MGS and DLC contribute to network convergence and optimization results.

## 6 Limitations and Broader Impact

Firstly, some of the other convolutions (*e.g.*, sparse convolution, group convolution, dilated convolution) and additional computer vision tasks remain unexplored. Secondly, the inference speed of the $\ell_1$-norm Net is marginally slower than that of traditional one. This is attributed to the lack of CUDA and cuDNN optimized operations for Manhattan distance metrics. It's noteworthy that, beyond introducing a novel convolution based on the $\ell_p$-norm and proving the universal approximation theorem for theoretical support, this paper also presents customized optimization strategies.

## 7 Conclusion

In this paper, we are motivated to explore $\ell_p$-norm measure to replace the classic inner product convolution. we first prove the universal approximation of $\ell_p$-norm Nets. And then we compare different $\ell_p$-norm measures and propose the $\ell_1$-norm Net for 3D point cloud tasks. Furthermore, we design the customized optimization strategies (*i.e.*, mixed gradient strategy and dynamic control on learning rate) for $\ell_1$-norm Net. When introducing our method to classical 3D networks, they achieve competitive performances at a lower energy cost. In summary, our $\ell_1$-norm Net can achieve similar performance to traditional convolution network, but with less computational cost and lower instruction latency.

## Acknowledgements

This work was supported in part by National Natural Science Foundation of China under Grant 62302143 and Anhui Provincial Natural Science Foundation under Grant 2308085QF207. Thanks for the help of Xinyuan Song.

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

# – Appendix –

In the Appendix, we present additional information on our methods. Concretely, we provides a detailed theoretical analysis of the theorems from the main paper, including the variance analysis, the proposed universal approximation, the regret argument, and the equivalence of $\ell_2$-norm Measure.

## A   Additional Theoretical Analysis

### A.1   Omitted proof of variance analysis

Since adding a constant does not significantly affect variance, for ease of demonstration, we could assume $Var\big[\|G + P_t - K\|_2\big] \approx Var\big[\|G\|_2\big]$. Notice that

$$Var\big[\|G\|_2\big] = \mathbb{E}_{G \sim N(0, \mathbb{I}_m)}\big[\|G\|_2^2\big] - \Big(\mathbb{E}_{G \sim N(0, \mathbb{I}_m)}\big[\|G\|_2\big]\Big)^2. \tag{16}$$

It's easy to verify that for any $u \geq 0$,

$$\sqrt{u} \geq (1 + u - (u-1)^2)/2.$$

Let $u = \frac{\|G\|_2^2}{m}$ and calculate the expectation of $G$ on both sides of the inequality, we have

$$\frac{\mathbb{E}[\|G\|_2]}{\sqrt{m}} \geq \frac{1}{2} \cdot \Big(2 - \mathbb{E}\Big[(\frac{\|G\|_2^2}{m} - 1)^2\Big]\Big). \tag{17}$$

Because $\mathbb{E}\big[(\frac{\|G\|_2^2}{m} - 1)^2\big] = \frac{1}{m^2} \cdot \mathbb{E}\big[\sum_{i=1}^m (G(i)^2 - 1)^2 + \sum_{i \neq j}(G(i)^2 - 1)(G(j)^2 - 1)\big]$ and $\forall i$, $\mathbb{E}[G(i)^2 - 1] = 0$, we can conclude that

$$\mathbb{E}\big[(\frac{\|G\|_2^2}{m} - 1)^2\big] = \frac{1}{m} \cdot \mathbb{E}[G(1)^4 + 1 - 2 \cdot G(1)^2] \tag{18a}$$

$$= \frac{2}{m}. \tag{18b}$$

where the first equation holds for all the $G(i)$s are i.i.d. The second equation holds for $\mathbb{E}[G(1)^4] = 3$, $\mathbb{E}[G(1)^2] = 1$. Combining inequality (17) and Equation (18b), $\mathbb{E}_{G \sim \mathcal{N}(0, \mathbb{I}_m)}[\|G\|_2] \geq \frac{\sqrt{m}}{2} \cdot (2 - \frac{2}{m})$. Therefore, by Equation (16) we have

$$Var[\|G\|_2] < 2 - \frac{1}{m} = O(1).$$

Thus we have shown that $Var\big[\|(G + P_t) - K\|_2\big] = O(1)$.

### A.2   Proof of Theorem 1

Scaling $S$ and $J$ by $\frac{1}{diam(J)}$ where $diam(J) = \max_{x,y \in J}\{\|x - y\|_\infty\}$, we could assume $S = \{x_1, \cdots, x_N\}$ and $\forall i \in [N]$ $x_i \in J \subset [0,1]^k$. For convenience, first we show the case $k = 1$. Here we refer the construction of soft occupancy function in [51]. Because $f$ is continuous function, for any $\epsilon > 0$, $\exists \sigma > 0$ so that $|f(S_1) - f(S_2)| < \epsilon$ for any $S_1$ and $S_2$ with $d_H(S_1, S_2) < \delta$. Let $M = \lceil \frac{1}{\delta} \rceil$ and $h_m(x) = exp(-d_H(x, [\frac{m-1}{M}, \frac{m}{M}]))$ be the soft occupancy function, for all $m \in [M]$. Next, for all $m \in [M]$ define

$$\hat{v}_m(S) = \max_{x \in S}\{h_m(x)\} \tag{19}$$

and,

$$v_m(S) = \begin{cases} 1, & \hat{v}_m(S) \geq 1 \\ 0, & \hat{v}_m(S) < 1. \end{cases} \tag{20}$$

$v_m(S)$ indicates the occupancy of the $m$-th interval by points in $S$. Define $\mathbf{v} : 2^J \to \{0, 1\}^M$ and for any $S \in 2^J$, $\mathbf{v}(S) = (v_1(S), v_2(S), \cdots, v_M(S))$. And then define $\eta : \{0, 1\}^M \to 2^J$, $\eta(\mathbf{v}(S)) = \{\frac{m-1}{M} \mid v_m(S) \geq 1\}$. Notice that by this construction, $d_H(\eta(\mathbf{v}(S)), S) < \frac{1}{M} \leq \delta$. So let $\omega : \{0, 1\}^M \to \mathbb{R}$ and $\omega(\mathbf{v}) = f(\eta(\mathbf{v}))$, we have

$$|\omega(\mathbf{v}(S)) - f(S)| = |f(\eta(\mathbf{v}(S))) - f(S)| < \epsilon \tag{21}$$

The last inequality holds for the definition of Hausdorff distance and continuity of $f$. Here $\omega$ and $\{h_m\}_{m=1}^M$ could be made up of a multi-layer perceptron network [51]. $\{\hat{v}_m\}_{m=1}^M$ consist of a max pooling layer on $\{h_m\}_{m=1}^M$ and $\{v_m\}_{m=1}^M$ can be composed of a simple perceptron layer on $\{\hat{v}_m\}_{m=1}^M$, which compares $\hat{v}_m(S)$ and 1. For the general cases $k \geq 1$, it suffices to get the same conclusion by simply extending the 1 dimensional functions $h_m, \hat{v}_m, v_m$ to $k$ dimension. So there is a $\ell_p$-PointNet++ $\mathcal{P}$ that can approximate any continuous function $f$ on $2^J$.

We employ the RBF theory of [57] to give the second conclusion. For completeness, we restate it here.

**Theorem 3** ([57]). *The radio basis networks consist of a family of functions(RBF) noted by $S_K$:*

$$\sum_{i=1}^H a_i \cdot K(\frac{x - z_i}{\sigma})$$

*where $x \in \mathbb{R}^d$, $z_i \in \mathbb{R}^d$, $\sigma \in \mathbb{R}$, $H \in \mathcal{N}$. $S_K$ is dense in $\ell_1(\mathbb{R}^d)$, if $K$ satisfies: 1.integrable bounded, 2.$K$ is continuous almost everywhere, 3.$\int K(x)dx \neq 0$.*

It's clear that the $\ell_p$-norm $\|\cdot\|_p : \mathbb{R}^d \to \mathbb{R}$ satisfies all the three conditions on $K$. Besides, a large enough $\ell_p$ based convolution layer with a full connected layer could represent all the functions $\sum_{i=1}^H a_i \cdot \|(\frac{x-z_i}{\sigma})\|_p$. So for any $\ell_1$-integrable function $g$, there exists an $\ell_p$-PointNet++ $\mathcal{P}'$ such that for any $\epsilon > 0$, $\int |g(x) - \mathcal{P}'(x)|dx < \epsilon$.

## A.3 Proof of Theorem 2

Before proving, we restate an important result in online learning and we will use it in the following.

**Lemma 1** ([58]). *For any $Q \in \mathcal{S}_+^d$ and convex feasible set $\mathcal{F} \subset \mathbb{R}^d$, suppose $u_1 = \min_{x \in \mathcal{F}} \|Q^{1/2}(x - z_1)\|$ and $u_2 = \min_{x \in \mathcal{F}} \|Q^{1/2}(x - z_2)\|$ then we have $\|Q^{1/2}(u_1 - u_2)\| \leq \|Q^{1/2}(z_1 - z_2)\|$.*

Our proof framework is similar to that of [58]. Here is a standard argument in momnet method.

**Lemma 2.** *Suppose $m_t = \gamma m_{t-1} + (1 - \gamma)g_t$ with $m_0 = \mathbf{0}$ and $0 < \gamma < 1$. We have*

$$\sum_{t=1}^{T^*} \|m_t\|^2 \leq \sum_{t=1}^{T^*} \|g_t\|^2.$$

*Proof.* By Cauchy-Schwarz and Young's inequality, we have

$$\|m_t\|^2 \leq \gamma\|m_{t-1}\|^2 + (1 - \gamma)\|g_t\|^2.$$

Note that $m_0 = \mathbf{0}$,

$$\frac{\|m_t\|^2}{\gamma^t} \leq (1 - \gamma) \sum_{i=1}^t \|g_i\|^2 \gamma^{-i}.$$

So we have

$$\|m_t\|^2 \leq (1 - \gamma) \sum_{i=1}^t \|g_i\|^2 \gamma^{t-i}.$$

Take the summation on $t$ for both sides of the inequality, we have the conclusion. □

So we could begin to prove Theorem 2. Suppose $\{x_t\} \subset \mathbb{R}^n$. As the notation before, $x^\star = \arg\min_{x \in \mathcal{F}} \sum_{t=1}^{T^*} h_t(x)$ and $x_{t+1} = \Pi_{\mathcal{F}, \alpha_t^{-1/2}}(x_t - \alpha(t) \cdot m_t) = \min_{x \in \mathcal{F}} \|\alpha(t)^{-1/2} \cdot (x - (x_t - \alpha(t) \cdot m_t))\|$. By Lemma 1, $\|\alpha(t)^{-1/2} \cdot (x_{t+1} - x^\star)\|^2 \leq \|\alpha(t)^{-1/2} \cdot (x_t - \alpha(t) \cdot m_t - x^\star)\|^2 = \|\alpha(t)^{-1/2} \cdot (x_t - x^\star)\|^2 + \|\alpha(t)^{1/2} \cdot m_t\|^2 - 2\langle q_t m_{t-1} + (1 - q_t)g_t, x_t - x^\star \rangle$. Rearrange the inequality, we have

$$\langle g_t, x_t - x^\star \rangle \leq \frac{1}{2(1 - q_t)}\left[\|\alpha(t)^{-1/2} \cdot (x_t - x^\star)\|^2 - \|\alpha(t)^{-1/2} \cdot (x_{t+1} - x^\star)\|^2 + \|\alpha(t)^{1/2} \cdot m_t\|^2\right]$$
$$+ \frac{q_t}{2(1 - q_t)} \cdot \left(\|\alpha(t)^{1/2} \cdot m_{t-1}\|^2 + \|\alpha(t)^{-1/2} \cdot (x_t - x^\star)\|^2\right). \tag{22}$$

The second inequality holds for Cauchy-Schwarz inequality and for any $a, b \in \mathbb{R}$, $ab \leq \frac{a^2 + b^2}{2}$.

Because $\{h_t\}_{t=1}^{T^*}$ are convex functions:

$$R_{T^*} = \sum_{t=1}^{T^*} h_t(x_t) - h_t(x^\star) \leq \sum_{t=1}^{T^*} \langle g_t, x_t - x^\star \rangle$$

So we have

$$R_{T^*} \leq \underbrace{\sum_{t=1}^{T^*}\left[\frac{1}{2(1 - q_t)}\left[\|\alpha(t)^{-1/2} \cdot (x_t - x^\star)\|^2 - \|\alpha(t)^{-1/2} \cdot (x_{t+1} - x^\star)\|^2\right] + \frac{q_t}{2(1 - q_t)}\|\alpha(t)^{-1/2} \cdot (x_t - x^\star)\|^2\right]}_{A}$$
$$+ \underbrace{\sum_{t=1}^{T^*}\left[\frac{1}{2(1 - q_t)}\|\alpha(t)^{1/2} \cdot m_t\|^2 + \frac{q_t}{2(1 - q_t)}\|\alpha(t)^{1/2} \cdot m_{t-1}\|^2\right]}_{B}. \tag{23}$$

First we bound the part A:

$$\sum_{t=1}^{T^*}\left[\frac{1}{2(1 - q_t)}\left[\|\alpha(t)^{-1/2} \cdot (x_t - x^\star)\|^2 - \|\alpha(t)^{-1/2} \cdot (x_{t+1} - x^\star)\|^2\right] + \frac{q_t}{2(1 - q_t)}\|\alpha(t)^{-1/2} \cdot (x_t - x^\star)\|^2\right]$$
$$\leq \frac{1}{2(1 - q_1)}\left[\sum_{i=1}^{n} \alpha_1^{-1}(x_1(i) - x^\star(i))^2 + \sum_{t=2}^{T^*}\sum_{i=1}^{n}(\alpha_t^{-1} - \alpha(t-1)^{-1})(x_t(i) - x^\star(i))^2 + \sum_{t=1}^{T^*}\sum_{i=1}^{n} q_t(x_t(i) - x^\star(i))^2 \alpha(t)^{-1}\right] \tag{24}$$

Next we bound the part B. By definition of $\alpha(t)$, $\alpha_1(1)/\sqrt{t} \leq \alpha(t) \leq \alpha_2(1)/\sqrt{t}$. So we have

$$\sum_{t=1}^{T^*}\left[\frac{1}{2(1 - q_t)}\|\alpha(t)^{1/2} \cdot m_t\|^2 + \frac{q_t}{2(1 - q_t)}\|\alpha(t)^{1/2} \cdot m_{t-1}\|^2\right]$$
$$\leq \frac{\alpha_2(1)}{2(1 - q_1)}\left[\sum_{t=1}^{T^*}\frac{\|m_t\|^2}{\sqrt{t}} + \sum_{t=1}^{T^*}\frac{\|m_{t-1}\|^2}{\sqrt{t}}\right]$$
$$\leq \frac{\alpha_2(1)}{2(1 - q_1)}\left[\frac{1}{T^*}\left[\sum_{t=1}^{T^*}\|m_t\|t^{-1/4}\right]^2 + \frac{1}{T^*}\left[\sum_{t=1}^{T^*}\|m_{t-1}\|t^{-1/4}\right]^2\right]$$
$$\leq \frac{\alpha_2(1)}{2(1 - q_1)}\left[\frac{1}{T^*}\sum_{t=1}^{T^*}\|m_t\|^2 \cdot \sum_{t=1}^{T^*}t^{-1/2} + \frac{1}{T^*}\sum_{t=1}^{T^*}\|m_{t-1}\|^2 \cdot \sum_{t=1}^{T^*}t^{-1/2}\right]$$
$$\leq \frac{\alpha_2(1)B_2^2}{(1 - q_1)}\sum_{t=1}^{T^*}t^{-1/2}$$
$$\leq (2\sqrt{T^*} - 1)\frac{\alpha_2(1)B_2^2}{(1 - q_1)}.$$

The second inequality holds for Jensen inequality and the third inequality follows from Cauchy-Schwarz inequality. The forth inequalityholds for Lemma 2.

Combine the argument above and notice that $\alpha(t)^{-1} \leq p_1^{-1} \cdot \sqrt{T^*}$, we have

$$
\begin{aligned}
R_{T^*} &\leq \frac{B_\infty^2}{2(1-q_1)} \left[ n \cdot \alpha_1^{-1} + \sum_{t=2}^{T^*} n \cdot (\alpha_t^{-1} - \alpha_{t-1}^{-1}) + \sum_{t=1}^{T^*} n \cdot q_t \cdot \alpha_t^{-1} \right] + (2\sqrt{T^*} - 1)\frac{\alpha_2(1)B_2^2}{1-q_1} \\
&\leq \sqrt{T^*} \cdot \left( \frac{B_\infty^2}{2(1-q_1)} \cdot n \cdot \hat{\alpha}_{T^*}^{-1} + \frac{2 \cdot \alpha_2(1)B_2^2}{1-q_1} \right) - \frac{\alpha_2(1)B_2^2}{1-q_1} + \frac{B_\infty^2}{2(1-q_1)} \sum_{t=1}^{T^*} n \cdot q_t \cdot \alpha_t^{-1} \\
&\leq \sqrt{T^*} \cdot \left( \frac{B_\infty^2 \cdot n \cdot p_1^{-1}}{2(1-q_1)} \cdot (1 + 2q_0 q) + \frac{2 \cdot \alpha_2(1)B_2^2}{1-q_1} \right) - \frac{\alpha_2(1)B_2^2}{1-q_1}.
\end{aligned}
$$

## A.4 Equivalence of $\ell_2$-norm Measure and Classic Convolution in Convergence

We find that $\ell_2$-norm Net is the linear transformation to the inner product convolution network, here we give the detailed calculation.

The output of the $\ell_2$-norm Net in Eq. 25.

$$
Y_{\ell_2}(P_t, K) = \sqrt{\sum_{t \geq 1} \sum_{i,j} |P_t(i,j) - K(i,j)|^2} \tag{25}
$$

Therefore, we can express it as the following:

$$
\begin{aligned}
Y_{\ell_2}^2(P_t, K) &= \sum_{t \geq 1} \sum_{i,j} (P_t(i,j)^2 + K(i,j)^2 - 2P_t(i,j)K(i,j)) \\
&= \sum_{t \geq 1} \sum_{i,j} (P_t(i,j)^2 + K(i,j)^2) - \sum_{t \geq 1} \sum_{i,j} P_t(i,j)K(i,j) \\
&= \sum_{t \geq 1} \sum_{i,j} (P_t(i,j)^2 + K(i,j)^2) - 2Y_{CNN}(P_t, K).
\end{aligned} \tag{26}
$$

Notably, the term $\sum_{i,j} K(i,j)^2$ remains constant for each channel, and $\sum_{t \geq 1} \sum_{i,j} P_t(i,j)^2$ represents the square of $\ell_2$-norm of each input patch. If this term is invariant across patches, the $\ell_2$-norm Net's output can be regarded as a linear transformation of the CNNs' output.

