# OpenReview forum: "Rethinking 3D Convolution in $\ell_p$-norm Space"
_NeurIPS.cc/2024/Conference — NeurIPS 2024 spotlight_

### Official Review · Reviewer_VAmn · 2024-07-04

**Soundness:** 3
**Presentation:** 4
**Contribution:** 4
**Rating:** 7
**Confidence:** 5

**Summary:**

The paper proposes using the \( \ell_p \)-norm, specifically the \( \ell_1 \)-norm, to replace the classic squared \( \ell_2 \)-norm convolution in 3D tasks. The \( \ell_1 \)-norm kernel function relies on addition, reducing computational cost. Initial gradient implementation revealed insufficient gradient values. To address this, the authors gradually transition the significant gradient from \( \ell_2 \) to \( \ell_1 \) as training progresses and employ momentum updates and learning rate scheduling. By replacing traditional 3D networks with their \( \ell_1 \)-norm networks, the paper demonstrates competitive performance on ModelNet classification and S3DIS semantic segmentation tasks at significantly lower costs.

**Strengths:**

i)  The proposed 3D \( \ell_1 \)-norm Net is novel and inspiring for 3D tasks,  which is broadly applicable somewhat.

ii)  The proposed tricks (the optimizer and the customized operator) make the \ell_1-norm network get competitive performance at a lower cost.

iii)  The proof of the appendix is detailed and carefully.

**Weaknesses:**

Minor typos error:

i) In L-651, it should be "The *\ell_p*-norm" rather than ""The \ell_p norm""

ii) Very few words require consistent capitalization.

iii) Formulas in separate rows may have missing order numbers, it is recommended to unify this style.

iv) The reference format is not uniform enough.

**Questions:**

i)  The current experiments seem to be on small datasets. Does the current method perform well on larger datasets?

ii)  During training, The key idea of this paper is to use the $\ell_2$ gradient. As far as I am concerned, another potential way is to use $\ell-_\infty$ gradients.  The dual space of $\ell__\infty$ is the $\ell_1$ norm; Thus, the gradients of $\ell_ _\infty$ will be sparse. Why did the authors not try this idea and speed up the training time?

iii)  The \ell-p-norm Space measurement can be also seen in other filed, such as  image analysis[1], Inpainting[2] , the author claim that "these method can't be **directly** applied into 3D tasks", why?

iv)  The proposed lower bound and upper bound are Eq(13) and Eq(14), why the authors design the bounds as these?



[1] Generalized 2-D principal component analysis by Lp-norm for image analysis.

[2] Rank-One Matrix Approximation With ℓ*p*-Norm for Image Inpainting.

**Limitations:**

More generalization experiments on other datasets can be considered.

---

> ### Author Rebuttal · Authors · 2024-08-06
>
> We thank the reviewer for their comments. We are very happy to receive such an enthusiastic review!
>
> # For Weaknesses:
> ***W1: Minor typos error***
>
> Thanks for your advice! We have carefully rechecked all the writing issues and will revise them in the next version, including consistent capitalization, formulas, reference format, etc.
>
> # For Questions:
> ***Q1: More experiments on larger datasets***
>
> Please refer to Sec.E.5.1, where we conduct the task *Garments pose estimation* using the baseline *GarmentNets*. In this task, the input is partial point clouds (incomplete and occluded point clouds), and the output is complete point clouds.
> This is a large-scale dataset, named GarmentNets Simulation, which contains six garment categories with a total data volume of *1.72TB*, including Dress, Jump, Skirt, Top, Pants, and Shirt.
>
> ***Q2: More experiments on larger datasets***
>
> $\ell _{\infty}$ norm isn't a good choice. Notice that for any  vector $y=(y(1),y(2),...,y(n))$,  $\Vert y \Vert _ {2}=\sqrt( \sum _ {i=1}^{n}y(i)^{2} )$ and $\Vert y \Vert _ {\infty}= max _ {1\le i \le n} |y(i)|$. So the gradient of $\Vert y \Vert _{2}$ is $ (y(1),y(2),y(3),...,y(n) )/(\Vert y \Vert _{2} )$ and the gradient of $\Vert y \Vert _{\infty}$ is $(0,0,0,..,0,sign(y(j)),0,....,0)$, where $j$ is the maximal entry of $y$ with respect to absolute value. Notice that there is only one non-zero entry in the gradient of $\Vert y \Vert _ {\infty}$ norm, which means only one dimension will be updated in each iteration. So $\ell _ {\infty}$ is very inefficient.
>
> In practical experiments, we find that it leads to a highly unstable training process easily, which can be corroborated by the results of Tab.2 (The quantitative result of  $\ell _{\infty}$ norm Net tells that it has suboptimal classification performance.)
>
> ***Q3: Why 2D methods can't be directly applied into 3D tasks***
>
> 3D point clouds and 2D images are inherently different forms of data organization, like spareness and disorderedness in 3D point clouds. Concretely, unlike an array of pixels in an image, a point cloud is a set of points in no particular order. In other words, a network that consumes $N$ 3D point sets needs to be invariant to the $N!$ ordering of the input sets in the order of data input, which extremely differs from the location-sensitive RGD domain. This leads to an undesirable result when the $\ell_p$-norm based method from the 2D tasks (such as image analysis and Inpainting) is used directly on the 3D tasks.
>
> ***Q4: About lower bound and upper bound Eq(13) and Eq(14)***
>
> As mentioned in L239-L244, we hope to achieve larger update magnitudes and faster convergence rates during the initial stages of training. To this end, a promising scheme for learning rate design is maintaining a higher rate in the early training phase, and returning to a lower rate in the later phase. Then we determine the lower bound and upper bound as Eq.(13) and Eq.(14), respectively.

---

> > ### Comment · Reviewer_VAmn · 2024-08-08
> > **Official Comment by Reviewer VAmn**
> >
> > Thanks for the clarifications. As the authors have clearly addressed my concerns, I'd like to increase my score ,and I'm happy to recommend it for acceptance.

---

### Official Review · Reviewer_4SD1 · 2024-07-04

**Soundness:** 4
**Presentation:** 3
**Contribution:** 4
**Rating:** 8
**Confidence:** 4

**Summary:**

The paper proposes for using the $\ell_{p}$ norm in 3D convolution as a substitute for the inner product. Out of different choices, the authors pick the $\ell_{1}$-norm because it's faster and uses less energy than the inner product. This is because the $\ell_{1}$-norm relies on addition, which is simpler computationally. To optimize the $\ell_{1}$-norm-based convolution models, they propose a tailored optimization approach leveraging Mixed Gradient Strategy and Dynamic Learning Rate Controller. Finally, the method is evaluated on ModelNet10 and ModelNet40 datasets, showcasing its usage in learning point cloud features for object classification.

**Strengths:**

1 ) The writing and organization of this paper is great.

2 ) The experiments are sufficient in both main paper and supp, ranging from global Tasks, Semi-dense Prediction, Dense Prediction.

3 ) The idea of replacing traditional convolution with a ℓ𝑝-based convolution is quite foundational and novel for 3D tasks.

4 ) The proposed methods can be easily extended to other 3D backbones, which can be more environmentally friendly.

5 ) The proposed methods demonstrate good performance on most tasks (such as ModelNet classification and S3DIS semantic segmentation) at a significantly lower cost.

**Weaknesses:**

1 )  The baseline models should be detailed introduced, including the model structure, integration (i.e., replacing) method of the proposed method, just as Table 8 from Ablation Experiments.

2 ) The layout of Table 5 needs to be further optimized, such as splitting it into two independent tables for parallel layout.

**Questions:**

1 )  Is there any basis for the authors to assume that the mean of data **X** is 0 when discussing robustness?

2 )  The authors discuss regret without detailed motivation, How does it works? I wonder if it is necessary to introduce the regret.

3 ) There are many symbols in this article, it is recommended to add more explanations  in a separate section.

**Limitations:**

See weakness and questions.

---

> ### Author Rebuttal · Authors · 2024-08-05
>
> Thank you for your constructive comments. Answers to some of your questions are as follows.
> # For Weaknesses
>
> ***W1: About baselines***
>
> For the Classification and Segmentation task, we use PointNet and PointNet++ as the baselines. PointNet processes point clouds by embedding each point independently using a shared MLP and then aggregating global features through max pooling.  PointNet++ extends this approach by incorporating hierarchical feature learning, grouping points into local neighborhoods and applying PointNet-like operations at multiple scales.
>
> For the Completion task, we use OccNet, DMC, and IFNets as the baselines. OccNet uses the network to define a continuous occupancy function, predicting the probability of a point being inside the object, instead of using discrete voxel grids. DMC extracts a triangulated mesh to provide the final surface reconstruction.  IF-Nets captures fine details through a hierarchical architecture, learning features at multiple scales for accurate reconstruction of complex geometries.
>
> In these experiments, traditional conventions are all replaced, rather than partial replacement as Tab.8.
>
> ***W2: About layout***
>
> Thanks for your advice! In the revised version, we will split it into two parallel tables, each with 8 different shapes.
>
> # For Questions
>
> ***Q1: About data $X$***
>
> The assumption that the mean of the data $ X $ is zero was made primarily to simplify the mathematical derivations in our analysis. Given that $ G $ represents random noise, the difference in variance between $ G $ and $ X + G $ becomes negligible when $ X $ is considered constant. This simplification does not impact the demonstration of the robustness properties of the $ \ell_2 $-norm, particularly in the context of convolution operations.
>
> Our focus was on evaluating the robustness of the $ \ell_2 $-norm under noise conditions, and the zero mean assumption for $ X $ allowed us to streamline the proofs without loss of generality. In practical scenarios, even if $ X $ has a non-zero mean, the robustness properties of the $ \ell_2 $-norm would still hold due to its inherent nature of minimizing the effect of noise.
>
> ***Q2: About the regret***
>
> Our standpoint is that regret is a valuable addition to our work, especially in the context of analyzing and proving the convergence properties of the optimization process. Regret, defined as the difference between the actual performance of an algorithm and the best possible performance in hindsight, provides a comprehensive measure of the efficiency and effectiveness of the learning or optimization algorithm.
>
> Introducing regret into our analysis has several advantages, for example, 1) Convergence Analysis. By analyzing the regret, we can demonstrate that the optimization process converges over time, thus providing a rigorous justification for the algorithm's performance guarantees. 2) Flexibility in Stochastic Environments.
> In scenarios involving uncertainty or noise, regret analysis helps in understanding the robustness and adaptability of the algorithm. It provides insights into how the algorithm performs under varying conditions and helps in identifying potential areas for improvement.
>
> In the appendix, we demonstrate how the regret bounds are derived and how they relate to the convergence of our proposed algorithm. By doing so, we offer a more robust theoretical foundation and validate the efficacy of our approach.
>
> ***Q3: About Notations***
>
> For a clearer expression, we have reiterated the notations in L191-L192.  In the revised version, we will add an additional section called 'Notations' for readers' reference before Sec.3 *Methodology*, Thanks for your advice!

---

> > ### Comment · Reviewer_4SD1 · 2024-08-10
> > **Official Comment**
> >
> > I appreciate the authors addressing my concerns.
> > The analysis presented in the Q1(the mean of data $X$ can be 0) and Q2 are valuable, as it allows the readers to understand that the theory and motivation more clearly. Now I do not have further questions for the authors. I agree that the current work is significant enough to justify acceptance.

---

### Official Review · Reviewer_uudN · 2024-07-10

**Soundness:** 3
**Presentation:** 2
**Contribution:** 3
**Rating:** 5
**Confidence:** 3

**Summary:**

This paper addresses the challenge of enhancing the representational capacity of traditional convolution methods. To tackle this issue, the authors introduce a novel convolution approach based on $\ell_p$-norm and offer customized optimization strategies to expedite the training process. Extensive theoretical and empirical results have verified that the proposed algorithms demonstrate competitive performance compared to other baselines.

**Strengths:**

The concept of using the $\ell_p$ norm-based kernel to develop a convolution operator is intriguing. This proposed method has the potential to notably enhance the flexibility of the traditional convolution operator.

**Weaknesses:**

1. The writing of this paper requires improvement, particularly in providing additional details regarding the proposed method. The current version of this manuscript may lead to reader confusion.

2. The role of the theoretical results presented in this paper is unclear. There appears to be a gap between the theoretical guarantees and the empirical evidence provided.

**Questions:**

1. The authors ought to include a formal definition of the "$\ell_p$ norm space" concept in the title of this paper.

2. The formulation of the proposed convolution operator in Eq. 2 on the 35th line of page 2 lacks an intuitive explanation, which may confuse the reader.

3. What precisely is the definition of $\mathcal{P}(S)$ in Eq. 3? Additionally, the primary method proposed in this paper is only briefly introduced in the introduction without further detailed description, making it less reader-friendly.

4. It appears that the proposed training strategy "Dynamic Learning rate Controller" is seemingly unrelated to the $\ell_p$ convolution method, if I haven't overlooked anything.

5. There are several errors are evident in theorem 2. Initially, the mismatch between the subscripts $k$ and $t$ in the statement "we could show that for any convex functions $\lbrace h_k \rbrace_{t=1}^{T^*}$ " is notable. Furthermore, it would be beneficial if the authors could clarify the significance of $h$. Moreover, although this theorem addresses convex functions, it is crucial to acknowledge that numerous deep learning tasks entail non-convex functions. In conclusion, the reasoning behind the authors' choice to include a regret analysis in theorem 2 within this paper lacks clarity.

6. Building on the previous point, it is advisable for the authors to undertake a conventional convergence analysis for the proposed optimization algorithms. This recommendation stems from the availability of numerous well-established mathematical tools designed for assessing convergence in offline optimization, particularly under non-convex or unbounded gradient conditions. These tools are often better suited for the intricacies of deep learning environments.

7. In the experimental section, it is recommended that the authors provide a comprehensive outline of the setup for the online learning tasks they consider, particularly detailing how data arrives sequentially. Additionally, the experimental section overlooks sequential decision tasks or time series tasks. Therefore, I suggest that the authors include these related tasks to further strengthen the justification for providing a regret guarantee in theorem 2. Besides, as this paper is affiliated with the learning theory track, the empirical results presented seem somewhat disconnected from the theoretical findings. Specifically, while theorem 2 offers a regret guarantee for the proposed method, the metric used in the experimental section does not measure regret.

8. In comparing the proposed method with baseline methods, the authors appear to have neglected to consider recent online learning methods, such as online non-convex learning using follow-the-perturbed-leader [1,2].

[1] Suggala, Arun Sai, and Praneeth Netrapalli. "Online non-convex learning: Following the perturbed leader is optimal." Algorithmic Learning Theory. PMLR, 2020.

[2] Suggala, Arun, and Praneeth Netrapalli. "Follow the perturbed leader: Optimism and fast parallel algorithms for smooth minimax games." Advances in Neural Information Processing Systems 33 (2020): 22316-22326.

**Limitations:**

The authors have adequately addressed the limitations and potential negative societal impact of their work.

---

> ### Author Rebuttal · Authors · 2024-08-06
>
> Thanks for the kind and constructive comments! Please see below for responses.
> # For Weaknesses
> ***W1: About additional details***
>
> We will add more details from the appendix in the revised version.
>
> ***W2: About the gap between theoretical guarantees and the empirical evidence***
>
> Due to the space limitation, we provide a clearer correspondence between the *main* theoretical contributions and experiments below
>
> 1. Universal Approximation Theorem ensures that different $\ell_p$ Nets can converge and extract features effectively. Corresponding empirical evidence is given in Tab.1 ($\ell_p$ Nets with different $p$ values all have certain classification ability.)
>
> 2. For robustness, the corresponding experiments in Section E.7 illustrate the performance under different noises.
>
> 3. By introducing regret, the network integrated with the proposed optimization strategy is proven to retain the convergence properties. The corresponding empirical results can be found in global tasks, semi-dense predictions, dense predictions, and ablation results (Tab.9).
>
> # For Questions
> ***Q1: About the formal definition***
>
> The concept of $\ell_p$ norm space is not entirely comprehensive but "3D Convolution in $\ell$p-norm Space". We have provided its formal definition in lines 34-36, including the formulaic expressions (Eq2) and conceptual diagrams (Fig1).
>
> ***Q2: About Eq. 2***
>
> The variables are first defined in Eq.1 (refer to L20-L25) and are used consistently throughout the paper. This approach minimizes redundancy and maintains narrative flow.
>
> ***Q3: About primary method***
>
> 1) As described in L106-L113, $S$ is the Input data, while $P(\cdot)$ is the proposed $\ell_p$-PointNet++, therefore, $P(S)$ is the extracted features (output).
>
> 2) The primary method of this paper is $\ell-p$-norm Nets and Optimization strategy, which is introduced in Sec.3.3 and  Sec.4, respectively.
>
> ***Q4: About training strategy***
>
> This strategy is independent of the convolution method's *design*; instead, it is specifically tailored for *optimizing the training* of the convolution method, just as described in L239-L244.
>
> ***Q5: About theorem 2***
> 1) The correct definition should be ${ h_t }_ {t=1}^{T^*}$, so as the following Equation.
>
> 2) $h$ represents the objective function that needs to be optimized at the time $t$.
>
> 3) As described in L174 - L175, regret helps to analyze and prove the convergence properties of the optimization process.
> $h$ in regret analysis quantifies the cost or penalty incurred when a state $x_t$ is taken at step $t$. The objective is to minimize the total loss over time, leading to better long-term outcomes or global minimizers. For non-convex optimization problems, although we promise $h$ to be convex, the total loss function $\sum_t h_t$ may have multiple local minima and saddle points. And regret-based online learning optimization method can be well applied to such non-convex optimization problems. Besides, since numerous commonly used loss functions are either convex or can be approximated by convex functions, our theoretical results offer valuable insights and practical guidance.
>
> ***Q6: About conventional convergence analysis***
>
> We provide the proof of the convergence theorem using conventional concentration inequalities under convex function conditions (refer to Appendix C for notation definitions):
>
> Let  $( \mathcal{F} \subseteq \mathbb{R}^n )$ be a convex feasible set, and $f \in \mathcal{F}$, we have:
>
> $f(\mathbf{x}_t)\leq f(\mathbf{x}^*)+\nabla f(\mathbf{x}_t)^T(\mathbf{x}_t-\mathbf{x}^*)$
>
> Next, by calculating and combining the Eq.25 in Appendix C, for some large $t = T$, we have:
>
> $T\cdot(f(\mathbf{x}_t)-f(\mathbf{x}^*))\leq\sum(f(\mathbf{x}_t)-f(\mathbf{x}^*))$
>
> $\leq\sum_t\left(\langle g_t,x_t-x*\rangle\right)$
>
> $=\sqrt{T}\cdot\left(\frac{B_{\infty}^{2}\cdot n\cdot p_{1}^{-1}}{2(1-q_{1})}\cdot(1+2q_{0}q)+\frac{2\cdot\alpha_{2}(1)B_{2}^{2}}{1-q_{1}}\right)-\frac{\alpha_{2}(1)B_{2}^{2}}{1-q_{1}}$
>
> Thus, we obtain:
>
> $\Vert f(\mathbf{x} _ t)-f(\mathbf{x}^*)\Vert \leq\frac{1}{\sqrt{T}}\cdot\left(\frac{B_\infty^2\cdot n\cdot p_1^{-1}}{2(1-q_1)}\cdot(1+2q_0q)+\frac{2\cdot\alpha_2(1)B_2^2}{1-q_1}\right)=O(\frac{1}{\sqrt{T}})$
>
> Besides, for any function that can be represented by summation of some convex function $\sum_N \left( f_N(x) \right), \mathcal{F}_N \subseteq \mathbb{R}^n$, our proof still holds. This is also a proof where the regret method can be applied. However, for some non-convex $f$ that can not be represented, this proof may not hold. Nevertheless, under the online learning regret framework in our paper, it can still assist us in understanding the convergence.
>
> ***Q7: Suggestions on measuring regret***
>
> Thank you for your advice. In the revised version, we will provide a comprehensive outline including the details of data stream, the incremental processing of each data point at time $t$, and the continuous model updates.
> ﻿
> More importantly, to achieve greater consistency between theory and experiments, we will include additional evaluations that specifically measure regret in the revised version. Concretely, we will implement a dedicated evaluation framework to compute and analyze the cumulative regret incurred. By doing so, we aim to empirically validate the regret bound established in Theorem 2 and provide a more explicit connection between theoretical guarantees and observed performance. Furthermore, we will explore more regret metrics, such as instantaneous regret and average regret.
>
> ***Q8: More comparisons***
>
> The experimental setting in this paper is based on offline learning algorithms, which have less strict convergence requirements than online learning methods. Hence, the experimental setting in this paper is based on offline learning algorithms. Since online and offline algorithms are under different task settings, direct comparison is unfair. In future work, we plan to standardize all baseline methods to an online setting and conduct comparative experiments.

---

> > ### Comment · Reviewer_uudN · 2024-08-11
> >
> > I appreciate the authors for addressing most of my concerns. However, I am still confused about the choice to conduct the theoretical analysis on a convex assumption, especially considering the well-established body of analysis that relies solely on smoothness for non-convex optimization. Therefore, I will maintain my current score.

---

> > > ### Author Response · Authors · 2024-08-12
> > > **Official Comment by Authors of Paper 1528.**
> > >
> > > Thank you for your reply. Regarding the issue you are concerned about, we will further clarify it below:
> > >
> > > ----
> > >
> > > ① Firstly, non-convex functions present significant challenges in theoretical analysis due to the limited tools available for studying their behavior. However, since most commonly used loss functions are either convex or can be approximated by convex functions, our theoretical results offer valuable insights and practical guidance.
> > >
> > > ----
> > > ② Here, we also provide the proof of the convergence theorem using conventional concentration inequalities under ***non-convex function*** and Hölder continuous assumption conditions (refer to Appendix C for notation definitions):
> > >
> > > **Assumption 1**. Let $\alpha \in (0, 1]$ and $L > 0$, we assume that the gradient of $f(x)$ is $\alpha$-Hölder continuous in the sense that:
> > >
> > > $\begin{aligned}\Vert \nabla f(x_l)-\nabla f(x^*) \Vert \leq L \Vert x_l-x^* \Vert _2^\alpha,\quad\forall x_l,x^*\in\mathbb{R}^n\end{aligned}$
> > >
> > > As demonstrated in [1], for any function $f(x): \mathbb{R}^n \mapsto \mathbb{R}$ with Hölder continuous gradients, we have the popular lemma playing an important role in our analysis. Our proof based on our algorithms provides a quantitative measure on the accuracy of approximating $f(x)$ with its first-order approximation.
> > >
> > > **Lemma 1.** Let $ f : \mathbb{R}^n \mapsto \mathbb{R} $ be a differentiable function. Let $ \alpha \in (0, 1] $ and $ L > 0 $. If for all $ x_t, x^* \in \mathbb{R}^n$
> > >
> > > $\Vert \nabla f(x_t)-\nabla f(x^*) \Vert _2\leq L \Vert x_t-x^* \Vert _2^\alpha,$
> > >
> > > then, we have
> > >
> > > $f(x^*) - f(x_t) \leq \langle x^* - x_t, \nabla f(x_t) \rangle + \frac{L}{1 + \alpha} \Vert x^* - x_t \Vert _2^{1 + \alpha}.$
> > >
> > > Let  $( \mathcal{F} \subseteq \mathbb{R}^n )$ be a feasible set satisfying Assumption 1, and $f \in \mathcal{F}$, we have:
> > > $$f(x^*) - f(x_t) \leq \langle x^* - x_t, g_t \rangle + \frac{L}{1 + \alpha} \Vert x^* - x_t \Vert _2^{1 + \alpha}$$
> > > Next, by calculating and combining equation 25 in Appendix C, since  $m_t$  is gradually approaching the optimal  $m^*$, it is reasonable to believe that  $x_t$  is also approaching the optimal  $x^*$. So for some large $t = T$, we have:
> > >
> > > $T\cdot(f(\mathbf{x} _ t)-f(\mathbf{x}^*))\leq\sum _ {t=1}^T\left(\langle g _ t,x _ t-x*\rangle\right)+\sum _ {t=1}^T\frac{L}{1+\alpha} \Vert x^*-x _ t \Vert _ 2^{1+\alpha}$
> > >
> > > $=\sqrt{T}\cdot\left(\frac{B_{\infty}^{2}\cdot n\cdot p_{1}^{-1}}{2(1-q_{1})}\cdot(1+2q_{0}q)+\frac{2\cdot\alpha_{2}(1)B_{2}^{2}}{1-q_{1}}\right) -\frac{\alpha_{2}(1)B_{2}^{2}}{1-q_{1}}+\frac{L}{1+\alpha}\sum_{t=1}^{T}\left(\sum_{i=1}^{n}\alpha(t)^{1/2}(x^{*}(i)-x_{t}(i))\right)^{2+2\alpha}$
> > >
> > > Combine the argument above and notice that $\alpha(t)^{-1} \leq p^{-1}_1 \cdot \sqrt{T}$, we have:
> > >
> > > $\Vert f(\mathbf{x} _ {t})-f(\mathbf{x}^{*}) \Vert \leq\frac{1}{\sqrt{T}}\cdot\left(\frac{B _ {\infty}^{2}\cdot n\cdot p _ {1}^{-1}}{2(1-q_{1})}\cdot(1+2q _ {0}q)+\frac{2\cdot\alpha _ {2}(1)B _ {2}^{2}}{1-q _ {1}}+\frac{B _ {\infty}^{2}\cdot n^{2+2\alpha}\cdot L}{(1+\alpha)\cdot p _ {1}}\right)=O(\frac{1}{\sqrt{T}})$
> > >
> > > Therefore, for any $\alpha$-Hölder continuous functions, our proof of convergence still holds. This is also a proof where the regret method can be applied. However, for some singular non-convex $f$, this proof may not hold. Nevertheless, under the online learning regret framework in our paper, it can still assist us in understanding the convergence.
> > >
> > > # Reference
> > > [1] Unregularized online learning algorithms with general loss functions. *Applied and Computational Harmonic Analysis 2017*.
> > >
> > > ----
> > > ③ In practice,  the experiments demonstrate that the proposed network integrated with the proposed optimization strategy retains its convergence properties. This is supported by the Sec.5 and Sec.E.
> > >
> > > ----
> > > If our rebuttals do not address your concerns to some extent, we would be happy to have further discussions.
> > > Your feedback is an important reference for us to improve the quality of our paper, and we attach great importance to it. Thank you again for your time and effort. We look forward to your reply.

---

### Official Review · Reviewer_dtmU · 2024-07-10

**Soundness:** 3
**Presentation:** 4
**Contribution:** 3
**Rating:** 7
**Confidence:** 5

**Summary:**

This paper introduces a new convolution method based on the \( L_p \)-norm. The authors provide a theoretical foundation by proving the universal approximation theorem for \( L_p \)-norm networks and analyzing the robustness and feasibility of \( L_p \)-norms in 3D tasks. Several key findings are highlighted in this work:

1. \( L_\infty \)-norm convolution is prone to feature loss.
2. \( L_2 \)-norm convolution essentially performs a linear transformation in traditional CNNs.
3. \( L_1 \)-norm convolution is an economical and effective method for feature extraction.

To further enhance the capabilities of \( L_1 \)-norm based networks, the paper proposes a series of customized training and optimization strategies.

In the experimental section, the authors apply their methods to classical 3D networks such as PointNet and PointNet++, achieving competitive performance at a lower cost. In summary, the \( L_1 \)-norm network can achieve similar performance to traditional convolutional networks but with reduced computational cost and lower instruction latency.

**Strengths:**

-- This paper is clear and easy to follow, which is well organized.

-- The proof of the universal approximation theorem for Lp-norm Nets and the analysis of the robustness and feasibility of Lp-norms are interesting and beautiful.

-- The comparative results between different $\ell_{p}$-norm-based convolutions presented in this paper are valuable, offering a meaningful technical reference for further method design and subsequent research in 3D vision.

**Weaknesses:**

There are some weakness/concerns need to be discussed:

--Noise distribution is an interesting issue, but this paper only analyzes the impact of Gaussian noise when considering random noise. How do the authors solve other noise distribution conditions？

--In this paper, how to come up with the idea of the \( L_p \)-norm convolution  is not discussed in detail, although its mechanism is well  illustrated.

**Questions:**

--This paper claim that \( L_p \)-norm are more robust than that based on inner product, however, why only \( L_2 \)-norm  is proved? how about the other cases when $p$ differs?

--In Table 2, the results from \( L_2 \)-norm Net is also competitive, why not choose the \( L_2 \)-norm Net for the further study but  \( L_1 \)-norm?

--In Sec 3.1, the proposed Universal Approximation, I wonder why  the second part is presented through integration? this is confusing compared with other works such as  "Multilayer feedforward networks universal approximator"？

Overall, I have some questions about the theory and motivation above, which hopes to be answered.

---

> ### Author Rebuttal · Authors · 2024-08-05
>
> Thank you for the valuable comments that help us improve our work. The following is a careful response and explanation about the weaknesses and questions.
> # For Weaknesses
> ***W1: About Noise distribution.***
>
> In this work, we focused on Gaussian noise because it is the most prevalent type of noise in real-world scenarios. Specifically, Gaussian noise offers advantages such as its mathematical tractability and the central limit theorem's implications, which make it a common assumption in many practical applications.
>
> Here, we provide additional proof under any symmetric, independent and identically distributed (i.i.d.) noise with standard component variance $ \operatorname{Var}[{G(i)}] = 1$. Since our data is normalized to a standard interval, hence selecting the variance of noise to be 1 is appropriate. Without loss of generality, let $\mathbb{E}[G(i)] = 0$.
>
> According to the *delta method*, we have $ \operatorname{Var}[G(i)^2] = C^{\prime} < \infty$
>
> Now, assuming $\mathrm{Var} \Vert G+P_{t}-K \Vert_{2} \approx\mathrm{Var} \Vert \Vert G  \Vert_{2} \Vert  $, and $G$  follows some form of noise $G \sim p_{noise}(i.i.d.) $, we have:
>
> $\operatorname{Var}\big[ \Vert G \Vert_2 \big] = \mathbb{E}_{ G \sim p _ {noise}} ( \Vert G \Vert_2^2 )  - \big( \mathbb{E} _ {G \sim p _ {noise}} [ \Vert G \Vert_2 ] \big)^2$
>
> According to Equation 19 in the paper, we also have:
>
> $\mathbb{E}\left[\frac{\left\|G\right\|_2}{\sqrt{m}}\right]\geq\frac{1}{2}\cdot\left(2-\mathbb{E}\left[\left(\frac{\left\|G\right\|_2^2}{m}-1\right)^2\right]\right).$
>
> Considering
>
> $\mathbb{E}\left[\left(\frac{\left\|G\right\|_2^2}{m}-1\right)^2\right]=\left(\mathbb{E}\left[\frac{\left\|G\right\|_2^2}{m}\right]-1\right)^2+\mathrm{Var}\left(\frac{\left\|G\right\|_2^2}{m}\right)$
>
> $=\left(\frac{1}{m}\sum_i^m((\mathbb{E}[G(i)])^2+\text{Var}[G(i)])-1\right)^2+\frac{1}{m^2}\left(\sum_i^m\text{Var}[G(i)^2]\right)$
>
> $=\frac{C^{\prime}}m$
>
> Combining these inequalities and equations, we get
>
> $\mathbb{E}_{G\sim p _ {noise}}\left[\left\|G\right\|_2\right]\geq\frac{\sqrt{m}}{2}\cdot\left(2-\frac{C'}{m}\right).$
>
> Therefore, by the above inequality, we have:
>
> $\mathrm{Var}[\left\|G\right\|_2]<m-\frac{m}{4}\cdot\left(2-\frac{C'}{m}\right)^2=C'-\frac{{C'}^2}{4m}=O(1)$
>
> Thus we have shown that $\mathbf {Var}[\Vert G+P_{t}-K\Vert_{2}] = O(1)$ is a general case for any finite noise occurring in the nature.
>
>
> ***W2: About idea of proposed method.***
>
> Here, let's reiterate our motivation:
> Traditional 3D Convolution employs the inner product as the similarity measurement between the filter and the input feature. While the inner product is a common choice, it is not the only possible similarity measurement function. For instance, similarity measurement using $\ell_p$-norms has proven to be an effective strategy, widely utilized in 2D tasks (*as discussed in Related Work*). However, one of the key features of 3D point clouds is their sparseness. $\ell_p$-norm models are reported to have natural advantages in handling sparse data, as they can extract relevant information from sparse data within a larger receptive field. This automatic complexity reduction facilitates efficient algorithm training.
> **Despite these advantages, understanding the characterization of 3D convolution in $\ell_p$-norm space and proposing effective algorithms that incorporate the properties of 3D tasks have not yet been fully explored.**
>
> # For Questions
> ***Q1: About other cases when p differs.***
>
> Investigating the robustness of other $\ell_p$ -norms is indeed important, hence, we conducted additional numerical simulations to explore the behavior of different $\ell_p$ -norms as shown in Tab.1 of the main paper. Our findings suggest that as $p$ increases, the robustness of the norm in the presence of random noise improves. Specifically, we observed a consistent decrease in variance under random noise conditions with increasing $p$. This indicates that higher $p$ values tend to mitigate the impact of noise more effectively than lower ones, including the $\ell_2$-norm.
> The improvement in robustness can be attributed to the nature of the $\ell_p$ -norm, which places more emphasis on larger deviations as $p$ increases. This property helps in diminishing the influence of random noise, which typically manifests as smaller perturbations. Therefore, norms with larger $p$ values are inherently more resistant to such disturbances.
>
> ***Q2: Why not choose the $\ell_2$-norm Net for the further study.***
>
> As discussed in the appendix,  $\ell_2$-norm Nets can actually be considered a translation of traditional convolutions.
>
> ***Q3: About Universal Approximation.***
>
> The integration is employed to assess the approximation quality of the neural network in an averaged sense across the entire domain $J$. Specifically, by considering the integral of the absolute error between the neural network output and the true function value, we aim to measure the overall approximation capability of the network. In other words, if the integral of the error is small, it means that the error is small on the vast majority of possible data points.
>
> This approach aligns with the concept of the mean approximation error, where a lower integral value indicates that the neural network provides a good approximation to the target function $g$ on average. It offers a holistic perspective on the network's performance, ensuring that even if the error at some individual points might be non-negligible, the overall approximation remains satisfactory when considered over the entire set $J$.
>
> In contrast to pointwise approximation guarantees, the integral approach provides a more flexible and often more practical metric for assessing the quality of function approximation, especially in high-dimensional spaces. This method ensures that the neural network can adequately capture the function's behavior in a broader context, making it a valuable tool for evaluating the network's performance in real-world scenarios.

---

### Author Response · Authors · 2024-08-06

Dear Reviewers,

We greatly appreciate your time and effort in reviewing our paper. Your constructive comments and insightful perspectives have been invaluable to our team's research. We have carefully considered each of your concerns and have addressed them below.

Besides, We are encouraged by the comments of the reviewers on the **Novelty** (Reviewers 4SD1, VAmn), **Competitive performance** (Reviewer 4SD1), **A well-organized draft** (Reviewers dtmU, 4SD1), **Good theoretical proof** (Reviewers dtmU, VAmn), **Intriguing and flexible method** (Reviewer uudN), **Sufficient experiments** (Reviewer 4SD1), and **Extensibility** (Reviewer 4SD1).

We are sincerely grateful for your constructive criticism and valuable insights, which have significantly contributed to the refinement and improvement of our work. **We hope our team's study can inspire future research on 3D models with low power consumption and low instruction latency.**

Best regards,

Authors of Paper 1528.

---

### Decision · Program_Chairs · 2024-09-25

**Decision:**

Accept (spotlight)

**Comment:**

The  paper introduced a new convolution method based on ℓp-norm. For theoretical support, it proved the universal approximation theorem for ℓp-norm based convolution, and analyze the robustness and feasibility of ℓp norms in 3D point cloud tasks. The paper received the recommendations of 2XAccept, 1XStrong Accept and 1XBorderline Accept. Given the consistent recommendations, I think the paper could be accepted. The authors are requested to revise the paper as per the rebuttal.